# Probiotic Potential of a Novel Vitamin B2-Overproducing *Lactobacillus plantarum* Strain, HY7715, Isolated from Kimchi

**Joo-Yun Kim, Eun-Jung Choi, Jae-Ho Lee** **, Myeong-Seok Yoo, Keon Heo \*, Jae-Jung Shim and Jung-Lyoul Lee**

R & BD Center, hy Co., Ltd., 22, Giheungdanji-ro 24beon-gil, Giheung-gu, Yongin-si 17086, Korea;
monera@hy.co.kr (J.-Y.K.); cej8951@hy.co.kr (E.-J.C.); jhlee@hy.co.kr (J.-H.L.); audtmrl@hy.co.kr (M.-S.Y.);
jjshim@hy.co.kr (J.-J.S.); jlleesk@hy.co.kr (J.-L.L.)
* Correspondence: gunny@hy.co.kr; Tel.: +82-70-7835-5982

**Abstract:** Vitamin B2, also known as riboflavin, is essential for maintaining human health. The purpose of this study was to isolate novel lactic acid bacteria that overproduce vitamin B2 and to validate their potential as probiotics. In this study, *Lactobacillus plantarum* HY7715 (HY7715) was selected among lactic acid bacteria isolated from Kimchi. HY7715 showed a very high riboflavin-producing ability compared to the control strain due to the high expression of *ribA*, *ribB*, *ribC*, *ribH*, and *ribG* genes. HY7715 produced 34.5 ± 2.41 mg/L of riboflavin for 24 h without consuming riboflavin in the medium under optimal growth conditions. It was able to produce riboflavin in an in vitro model of the intestinal environment. In addition, when riboflavin deficiency was induced in mice through nutritional restriction, higher levels of riboflavin were detected in plasma and urine in the HY7715 administration group than in the control group. HY7715 showed high survival rate in simulated gastrointestinal conditions and had antibiotic resistance below the cutoff MIC value suggested by the European Food Safety Authority; moreover, it did not cause hemolysis. In conclusion, HY7715 could be considered a beneficial probiotic strain for human and animal applications, suggesting that it could be a new alternative to address riboflavin deficiency.

**Keywords:** riboflavin; vitamin deficiency; lactic acid bacteria; *rib* operon

## 1. Introduction

Riboflavin, also known as vitamin B2, is a water-soluble vitamin found in a variety of nutritional sources. It is not synthesized by the human body and must be obtained through consumption of a variety of animal and plant foods, including beans, meat, and dairy products [1]. The main active forms of vitamin B2 are flavin mononucleotide (FMN) and flavin adenine dinucleotide (FAD), which act as coenzymes in mitochondrial metabolism and as hydrogen carriers in biological redox reactions [2]. Most riboflavin is absorbed in the proximal small intestine and is mainly stored in the liver, spleen, kidney and heart muscles in the form of FAD [3,4]; the riboflavin that is not absorbed or stored is excreted in the urine [5]. The daily recommended intake (RDI) of riboflavin is 1.3 mg/d and 1.1 mg/d, respectively, for men and women, to maintain good health [3,6]. Riboflavin is continuously excreted in the urine, making its deficiency relatively common when the vitamin intake is insufficient [5,7]. Riboflavin deficiency may be related to malabsorptive conditions, including inadequate dietary intake and wasting disease. Symptoms associated with riboflavin deficiency (ariboflavinosis) include stomatitis, cheilosis, and genital rash of the mouth and tongue. It has also been associated with visual impairments, such as night blindness and psychological disorders, including migraine headaches, mild anemia, and depression [8–10]. Riboflavin deficiency is relatively rare in developed countries but remains a serious health problem in developing and industrialized countries as deficiency can occur in up to 50% of the elderly or young population [6,11].

To prevent riboflavin deficiency, many countries, including the United States, have included this vitamin in essential fortification programs in mass consumption foods. However,

vegetarians or people who rarely consume dairy products are still exposed to riboflavin deficiency [12–14]. Therefore, some of the beneficial microorganisms, including lactic acid bacteria (LAB), have been proposed as alternative means to increase the concentration of group B vitamins in foods based on their de novo biosynthetic ability [15,16]. Riboflavin-producing LAB have been obtained from various ecological environments, including dairy products, human gastrointestinal tract, grains, and fermented vegetables [15,17–20]. In addition, roseoflavin, a toxic analogue of riboflavin, is often used to induce spontaneous mutations to obtain LAB strains with high riboflavin productivity [19–24].

Several studies have reported on manufacturing foods with increased riboflavin, such as bread, pasta, and soy milk, using LAB that overproduces riboflavin [18–20], and reports on improving riboflavin deficiency using fermented soy milk rich in riboflavin are also available [16]. There have also been studies on the treatment of gastrointestinal diseases using riboflavin overproducing LAB [25–27]. However, there is not enough evidence on whether consuming these foods can resolve riboflavin deficiency, and there is no research on whether riboflavin is absorbed in vivo by directly ingesting riboflavin-overproducing LAB. Therefore, this study was conducted to discover a new LAB strain with high riboflavin productivity in order to develop beneficial probiotic strain that can be used as alternative vitamin B2 supplements. For this, a LAB strain with high riboflavin production capacity was selected using chemically defined medium (CDM) containing roseoflavin, and the molecular mechanism of riboflavin overproduction was verified through *rib* operon gene analysis. In addition, the potential of this LAB as a probiotic was confirmed through simulated gastrointestinal track (GIT), and the production of riboflavin in vivo was confirmed using riboflavin deficiency animal model. Finally, the biological safety of this strain was verified through antibiotic resistance test, hemolytic reaction test and whole-genome analysis.

## 2. Materials and Methods

### 2.1. Sample Collection and LAB Strains Isolation

Fifty fermented Kimchi samples were collected from 26 nationwide traditional markets in South Korea. Samples were obtained in duplicates using sterile tubes and stored at −80 °C until used in experiments with the addition of 50% glycerin (*v*/*w*). Kimchi samples were homogenized with a Polytron homogenizer. Homogenized samples were serially diluted with phosphate buffered saline (PBS) and plated on de Man, Rogosa and Sharpe (MRS) and M17 glucose agar plates (Difco, Sparks, MD, USA). The plates were incubated for 48 h in an anaerobic chamber (Coy Laboratory Products, Grass Lake, MI, USA) at 37 °C. Two hundred colonies were randomly selected from all MRS and M17 plates at a low dilution ($10^3$ to $10^4$). Selected colonies were grown in MRS and M17 glucose broth and streaked on agar plates to confirm purity.

### 2.2. Riboflavin Overproducing LAB Isolation and Identification

Single colonies of bacteria isolated from Kimchi were inoculated into riboflavin-free modified chemically defined medium (CDM) as mentioned in Table 1 and incubated at 37 °C for 24 h. The bacterial colonies (100 μL) were plated on riboflavin-free CDM agar plates and incubated at 37 °C for 48 h. Bacterial species identification was carried out by selecting only those plates that showed bacterial colonies in the riboflavin-free CDM. Thereafter, species identified as LAB were inoculated in riboflavin-free CDM broth, cultured for 24 h, and 100 μL of inoculum was plated on riboflavin-free CDM agar plate containing 150 μg/mL roseoflavin (Cayman Chemical, Ann Arbor, MI, USA). 16S rRNA gene sequencing was performed using universal rRNA gene primers (27F and 1492R) to identify isolated LAB strains and each process, including gene amplification, sequencing, and basic analysis, was carried out by Chunlab (Chunlab Inc., Seoul, Korea). 16S rRNA sequencing results were compared with the Genbank database through the Basic Local Alignment Search Tool (http://blast.ncbi.nlm.nih.gov/Blast.cgi, accessed on 21 June 2016) of the National Center

for Biotechnology Institute (NCBI). All vitamins, minerals, and amino acids constituting the CDM were purchased from Sigma Aldrich (Sigma-Aldrich, St. Louis, MO, USA).

**Table 1.** Composition of the chemically defined medium (CDM).

| Compound | Concentration (g/L) | Compound | Concentration (g/L) |
|---|---|---|---|
| Glucose | 20 | L-glutamine | 0.6 |
| $KH_2PO_4$ | 3.1 | L-leucine | 0.4 |
| $K_2HPO_4$ | 1.5 | L-alanine | 0.2 |
| $MgSO_4$ | 0.5 | L-lysine | 0.2 |
| $MnSO_4$ | 0.02 | L-phenylalanine | 0.2 |
| $FeSO_4$ | 0.02 | L-proline | 0.2 |
| Tween 80 | 1 | L-serine | 0.2 |
| Inosine | 0.1 | L-cysteine | 0.2 |
| Xanthine | 0.02 | L-arginine | 0.2 |
| Biotin | 0.0005 | L-asparagine | 0.2 |
| Cyanocobalamin | 0.0001 | L-tryptophan | 0.2 |
| Folic acid | 0.0001 | L-valine | 0.2 |
| *p*-Aminobenzoic acid | 0.0005 | L-histidine | 0.2 |
| Nicotinic acid | 0.0005 | L-glycine | 0.2 |
| Calcium pentothenate | 0.0005 | L-threonine | 0.2 |
| Pyridoxine | 0.0005 | L-isoleucine | 0.2 |
| Myo-inositol | 0.0005 | L-methionine | 0.1 |
| L-ascorbate | 0.0001 | L-tyrosine | 0.1 |

*2.3. Quantitative Analysis of Vitamin B2*

The concentration of vitamin B2, including riboflavin, FAD and FMN, was determined by high-performance liquid chromatography (HPLC) combined with fluorescence detector (Agilent-1100 Series, Palo Alto, CA, USA) based on previous studies with some modifications [19,21,28–30]. Riboflavin, FAD, and FMN were separated from culture medium or mouse plasma using a Unison UK-C18 column (4.6 × 100 mm, 3 μm). All samples were diluted 2–5 times in 50% methanol and then filtered through a 0.22 μm membrane filter (Millipore, Burlington, MA, USA). Methanol and 10 mM sodium dihydrogen phosphate ($Na_2HPO_4$) solution (pH 5.5) were used as the mobile phase. All solvents used in the mobile phase were HPLC grade and were degassed by ultra-sonication for 2 h. The flow rate was 0.8 mL/min, the injection volume was 20 μL, and the excitation and emission wavelengths were 445 and 530 nm, respectively. Riboflavin, FAD, and FMN standards Fluka (St. Gallen, Büchs, Switzerland) were used to compare the retention time and peak area of the sample to determine the contents of the analytical sample.

*2.4. Expression Analysis of Riboflavin Biosynthesis Genes*

The expression of the riboflavin operon genes required for riboflavin biosynthesis, including ribA, ribB, ribC, and ribG, was confirmed by real-time Polymerase chain reaction (PCR). LAB was cultured in riboflavin-free CDM or MRS broth at 37 °C for 24 h. Cultured LAB were centrifuged at $13,000 \times g$ for 5 min, and total RNA was isolated using the AllPrep bacterial RNA kit (QIAGEN Sciences Inc, Germantown, MD, USA) according to the manufacturer's instructions. USA cDNA was synthesized by Reverse Transcription System (QIAGEN Sciences Inc, Germantown, MD, USA). The synthesized cDNA was subjected to real-time PCR using PowerTrack SYBR Green Master Mix (Applied Biosystems, Foster City, CA, USA) and QuantStudio 6 Real-Time PCR system (Applied Biosystems, Foster City, CA, USA). The primer sequences of the riboflavin biosynthesis genes were designed using Primer-BLAST (www.ncbi.nlm.nih.gov/tools/primer-blast/, accessed on 28 December 2020) based on the genomic sequence of *Lactobacillus (L.) plantarum* JDM1 (NC_012984.1), which is known to have a complete riboflavin operon (Table 2). Bacteria universal 16S rRNA primer [31] and *Lactobacillus* specific 16S rRNA primer [32] were used as controls to normalize the expression level of each gene.

**Table 2.** Polymerase chain reaction (PCR) primer sequence of riboflavin biosynthesis genes.

| | Sequence | Tm (°C) | Reference |
|---|---|---|---|
| ribA | F: CGATGACTAGTGAACACGAT | 55.0 | |
| | R: CCCGGGATGATAGAAATCAG | 55.1 | |
| ribB | F: TTAGATGGGCACATTGTTCA | 55.0 | |
| | R: TAAACTGATCAACAGACGCA | 54.9 | NC_012984.1 |
| ribG | F: ATCCAATCGTGGGTGGTAAT | 53.4 | |
| | R: TATCGCCGTTTTTAGGGTGA | 53.8 | |
| ribH | F: TAAGATTGGGATTGTCGTGG | 55.1 | |
| | R: TAGCGCTTGGTTGTAATCAT | 55.1 | |

*2.5. Sequence Comparison of RFN Element Present Upstream of the Rib Operon*

*L. plantarum* HY7715 (HY7715) with the highest riboflavin production and *L. plantarum* NCDO 1752 (NCDO1752) were selected to identify a regulatory region (RFN element) upstream of the rib operon through DNA sequencing. Sequencing primers were designed based on the nucleotide sequence of NCDO1752 published in Genbank (DQ645592); the sense and antisense sequences of the primers were 5′-GCCACATTTTATTCAACTCCG-3′ and 5′-GGCTTGATCGATAGCCATTT-3′, respectively. DNA sequencing was performed by Chunlab, and the sequencing results were aligned using Vector NTI (Thermo Fisher, Waltham, MA, USA).

*2.6. Stability Study of HY7715 Riboflavin Overproduction Characteristics*

HY7715 and NCDO1752 were subcultured in CDM and MRS broth, respectively, at 37 °C for 30 generations and the level of riboflavin produced in the medium was measured every five generations to determine the stability of the riboflavin overproduction phenotype.

*2.7. Optimization of Media Composition for Improving Riboflavin Production*

The culture media for HY7715 was optimized to increase biomass and riboflavin production. Media optimization was performed by referring to existing studies [33–35], and the core components of media were determined through various trials as listed in Table 3. All ingredients were dissolved in tertiary distilled water (3DW) and then autoclaved at 121 °C for 15 min to prepare the culture medium. MRS pre-cultured HY7715 was inoculated (1%) in 10 mL of each prepared medium and incubated at 37 °C for 24 h. For each medium, after dilution in PBS and plating on MRS agar, viable HY7715 was quantified and riboflavin concentration was analyzed by the method described above (Section 2.3). Yeast extract and soy peptone were purchased from Difco and other ingredients were obtained from Sigma Aldrich.

**Table 3.** Media composition for HY7715 cultivation.

| Ingredients | Concentration (g/L) | | | |
|---|---|---|---|---|
| | Media 1 | Media 2 | Media 3 | Media 4 |
| Glucose | 20 | 20 | 20 | 20 |
| Yeast extract | 40 | 30 | 10 | 0 |
| Soy peptone | 0 | 10 | 30 | 40 |
| Sodium citrate | 5 | 5 | 5 | 5 |
| $K_2HPO_4$ | 2 | 2 | 2 | 2 |
| L-ascorbate | 0.05 | 0.05 | 0.05 | 0.05 |
| Tween 80 | 1 | 1 | 1 | 1 |
| $MgSO_4$ | 0.05 | 0.05 | 0.05 | 0.05 |
| $MnSO_4$ | 0.1 | 0.1 | 0.1 | 0.1 |
| L-cysteine | 0.01 | 0.01 | 0.01 | 0.01 |

*2.8. Optimization of Culture Conditions to Improve Riboflavin Production*

HY7715 was inoculated into 10 mL MRS and incubated at 37 °C. From the 5 mL MRS pre-culture, 1 mL was inoculated into a 500 mL flask containing 100 mL optimal medium, and

the culture was statically incubated at 37 °C for 16 h. The 2 L scale fermentation was carried out in a 3 L bioreactor (CNS, Daejeon, Korea); 20 mL of HY7715 culture solution obtained from 500 mL pre-culture was inoculated into 1.98 L optimal medium (pH and temperature-controlled fermentation). The starting pH was 6.0; when incubation was over, pH was varied using NaOH solution to keep the pH control at 4.5, 5.0, 5.5 or 6.0. The incubation temperature was set from 25 °C to 37 °C [36]. All batch fermentations were carried out for a total of 24 h and vitamin B2 (riboflavin, FAD, and FMN) was measured after fermentation. Mass fermentation (500 L) was performed under the optimal pH and culture temperature conditions determined above, and 100 mL of culture medium was collected every 3 h to measure the number of viable cells and analyze vitamin B2. Mass fermentation was carried out in hy probiotics plant (hy Co., Ltd., Pyeongtaek, Korea). The 2000 L fermenter used for mass cultivation was manufactured by CNS (CNS, Daejeon, Korea).

### 2.9. Measurement of Riboflavin Production in Co-Culture Conditions

The ability of HY7715 to overproduce riboflavin under co-culture conditions with Caco-2 cells was investigated as previously described [24,37]. The human colorectal Adenocarcinoma Caco-2 cell line was purchased from the American Type Culture Collection (ATCC, Manassas VA, USA). HY7715 and NCDO1752 were resuspended in antibiotics-free DMEM and incubated with Caco-2 (LAB:Caco-2 cells::100:1) at 37 °C in a humidified incubator containing 5% $CO_2$ for 8 h. Caco-2 cells cultured in DMEM without microbial inoculation were used as a control (untreated). After incubation, medium was collected and riboflavin was extracted and quantified by HPLC according to the method described in Section 2.3.

### 2.10. Prevention of Riboflavin Deficiency In Vivo

Specific pathogen-free (SPF) male BALB/c mice, weighing 20–22 g, were purchased from DooYeol Biotech (Seoul, Korea) and maintained in a testing facility in hy CO., Ltd. (Yongin, Korea) for 1 week prior to the experiment. Thereafter, mice were maintained in the SPF barrier chamber at controlled temperature (22 ± 2 °C), humidity (50 ± 10%), and 12:12-h light-dark cycle. Animals were fed sterilized pellet diet (Purina, Seoul, Korea) and sterilized water and were divided into 4 groups (Figure 1): fed AIN—93G (Dyets, Bethlehem, Palestine) control diet (C, *n* = 5); 7 d of control diet followed by AIN—93G_RD (Dyets Inc., Bethlehem, Palestine) riboflavin-deficient diet (D, *n* = 5); 7 d of control diet with NCDO1752 followed by riboflavin-deficient diet (NCDO1752, *n* = 5); 7 d of control diet with HY7715 followed by riboflavin-deficient diet (HY7715, *n* = 5). NCDO1752 and HY7715 suspended in PBS were administered orally once daily at a dose of $1.0 \times 10^9$ CFU/kg/mouse. The composition of the AIN-93G and AIN-93G RD diets were listed shown in Table S2. At the end of the experiment, animals were euthanized under carbon dioxide ($CO_2$) gas and then dissected to collect the contents of the cecum. Plasma was separated using an EDTA tube (BD, Franklin Lakes, NJ, USA) and riboflavin levels in plasma were analyzed by HPLC. Cecum contents were used to measure the rinG gene expressed by LAB in the intestine by performing real-time PCR after RNA isolation. Total RNA was isolated using the AllPrep PowerFecal RNA kit (QIAGEN Sciences Inc, Germantown, MD, USA), and cDNA synthesis and real-time PCR were performed as described in Section 2.4. All studies were conducted according to the hy Co., Ltd. animal experiment guide and approved by Institutional Animal Care and Use Committee of the hy Co., Ltd. (IACUC approval number, AEC-2021-00003-Y).

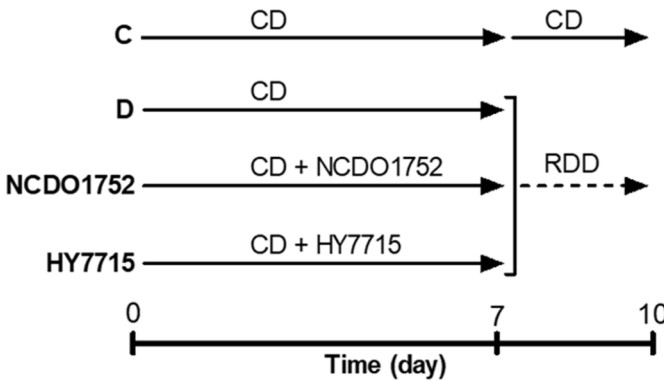

**Figure 1.** Timeline and treatment for in-vivo experiment.

*2.11. Intestinal Cell Adhesion Assay*

The rate of adhesion of HY7715 and NCDO1752 to Caco-2 cells, a human adenocarcinoma colon cell, was measured as previously described [24,38]. Caco-2 cells were cultured in Dulbecco's Modified Eagle Medium (DMEM) supplemented with 10% heat inactivated fetal bovine serum (FBS) and 1% antibiotic-antimycotic (Gibco) at 37 °C in an atmosphere containing 5% $CO_2$. Caco-2 cells were seeded at the concentration of $2 \times 10^5$ cells/well in 12-well plates. After the cells were grown to 100% confluence, the medium was replaced with DMEM without antibiotics and FBS. MRS cultured HY7715 and NCDO1752 were diluted in PBS, inoculated at a concentration of $1.0 \times 10^8$ CFU/mL in each well containing CaCo-2 cells, and incubated for 2 h at 37 °C in 5% $CO_2$ to adhere to CaCo-2 cells. After cultivation, the cells were washed 3 times with PBS and treated with 0.025% trypsin-EDTA (Gibco) for 10 min to separate the cells from the plate. The isolated bacterial cells were serially diluted with PBS and plated on MRS agar medium. Plates were incubated at 37 °C for 72 h for colony counting.

*2.12. Survivability in the Simulated Gastrointestinal Tract*

Survival rate was measured under physiological conditions similar to human gastrointestinal tract (GIT) to assess the potential of HY7715 as probiotics [39]. For this purpose, in vitro digestion tests were performed as a method to assess tolerance to saliva, gastric juices, and bile salts. Simulated saliva (SSF), simulated gastric fluid (SGF), and simulated intestinal fluid (SIF) were preheated to 37 °C prior to in vitro digestion. Then, 5 mL of HY7715 or NCDO1752 culture was poured in a 50 mL conical tube. The oral phase was performed by adding 26 μL of 0.3 M $CaCl_2$ solution and 4 mL of 6.55 mg/mL α-amylase solution in SSF. This was followed by the addition of 1 M NaOH, to adjust pH to 7.0, and incubation at 37 °C for 2 min. The gastric phase was performed by adding 6 μL of 0.3 mol/L $CaCl_2$, 694 μL of distilled water and 9.1 mL of 0.07 mg/mL pepsin in SGF. The pH was adjusted to 3 by adding 1 M HCl, with the mixture incubated at 37 °C by shaking continuously for 2 h. Finally, the intestinal phase was carried out by mixing 40 μL of 0.3 M $CaCl_2$, 1.31 mL of distilled water, 2.5 mL of 160 mM bile extract, and 16 mL of 22.15 mg/mL pancreatic solution in SIF. The pH was raised to 7.0 by adding 1 M NaOH, and the mixture was incubated for 2 h at 37 °C with shaking. At the end of each step, aliquots were collected and the viable cell count was measured to determine the digestive survival rate at each step.

*2.13. Safety Evaluation*

The minimum inhibitory concentrations (MIC, μg/mL) of eight antibiotics, including ampicillin, clindamycin, chloramphenicol, erythromycin, gentamicin, kanamycin, tetracycline and streptomycin, were determined according to the ISO 10932:2010 standard [35]. Epidemiologic cut-off values were based on recommendations from the European Antimi-

crobial Susceptibility Testing Committee (EUCAST) and European Food Safety Authority (EFSA) [40]. All antibiotics were purchased from Sigma Aldrich.

Hemolytic activity was measured by incubation for 48 h on Columbia agar containing 5% sheep blood (KisanBio, Seoul, Korea). The clear zone around the colony indicated hemolytic activity owing to the presence of β-hemolysin [35].

Gelatinase (coccolysin, gelE), hyaluronidase (hyl), aggregation substance (asa1), enterococcal surface protein (esp), cytolysin (cyIA), enterotoxin, non-hemolytic enterotoxin (nhe), hemolysin (hbl), cereulide (ces), serine protease (sprE), and transposon related (int, intTn) enzymes are known as potential virulence factors present in microorganisms [41]. Accordingly, HY7715 was analyzed for virulence-specific genes through whole-genome sequencing analysis, which was performed by Chunlab.

### 2.14. Statistical Analyses

Statistical results were presented as mean ± standard error of means (SEM) with 95% confidence limits. Data were compared statistically with the one-tailed one-way analysis of variance (ANOVA) plus a post hoc Tukey test. All statistical analyses were performed on Minitab (State College, PA, USA) statistical software. The *p* values less than 0.05 were considered statistically significant.

## 3. Results

### 3.1. Isolation of Riboflavin over Production LAB from Kimchi

Kimchi is a fermented vegetable dish made from cabbage or radish, with a high content of group B vitamins [42–44]. Various LABs, including *L. plantarum*, *L. fermentum*, *L. sakei*, *Leuconostoc (Lc.) mesenteroides*, *L. brevis*, *L. curvatus*, and *Lactococcus (Lc.) lactis* have been identified in Kimchi [45–47]. Therefore, in this study, 26 Kimchi samples collected from different regions of Korea were selected as raw materials for the screening of riboflavin-producing LAB.

About 250 LAB were isolated from Kimchi (data not shown), of which 46 formed colonies in the riboflavin-free CDM. As a result of 16S rRNA sequencing of 46 LAB, 33 *L. plantarum*, seven *Lc. lactis*, three *L. fermentum*, two *L. paracasei*, and one *L. farciminis* were identified. When LAB, which formed colonies on CDM plates, were inoculated into riboflavin-free CDM broth, only 24 strains grew. Among the LAB that could be grown in CDM, 18 strains produced riboflavin, and five had natural resistance to roseoflavin (Table S1). Strains that formed colonies in the riboflavin-free CDM containing 150 μg/mL of roseoflavin were four of *L. plantarum* (No. 2, No. 3, No. 9, and No. 14) and one of *L. fermentum* (No. 21). Colonies formed on CDM plates containing roseoflavin have been found to change the color of the medium from white to yellow during liquid culture in riboflavin-free CDM (Figure S1). This can be a qualitative indicator of riboflavin over-production [19]. After measuring the vitamin B2 content of No. 2, No. 3, No. 9, No. 14, and No. 21, No. 9 was found to produce the highest riboflavin at 1.85 ± 0.20 mg/L. There was no significant difference in FAD and FMN levels between strains (Figure 2A). To confirm that the riboflavin production characteristics of above strains were maintained even in the presence of riboflavin, strains were cultured in CDM (0.1 mg/L riboflavin contained) and MRS broths for 24 h, and then the amount of vitamin B2 (riboflavin and its cofactors) was measured. MRS is a medium that is commonly used for LAB culture and contains riboflavin [20,48]. Strain No. 9 produced high riboflavin even in the presence of riboflavin, which was in contrast to the remaining strains (Figure 2B). In addition, strain 9 produced 5.06 ± 2.05 mg/L of riboflavin in MRS broth, while the other strains produced little riboflavin in MRS broth (Figure 2C). LAB strain No. 9 showed the highest riboflavin production among the selected strains and was able to produce riboflavin even in a growing environment with riboflavin. Therefore, this strain was named *L. plantarum* HY7715 (HY7715), and the following experiment was performed.

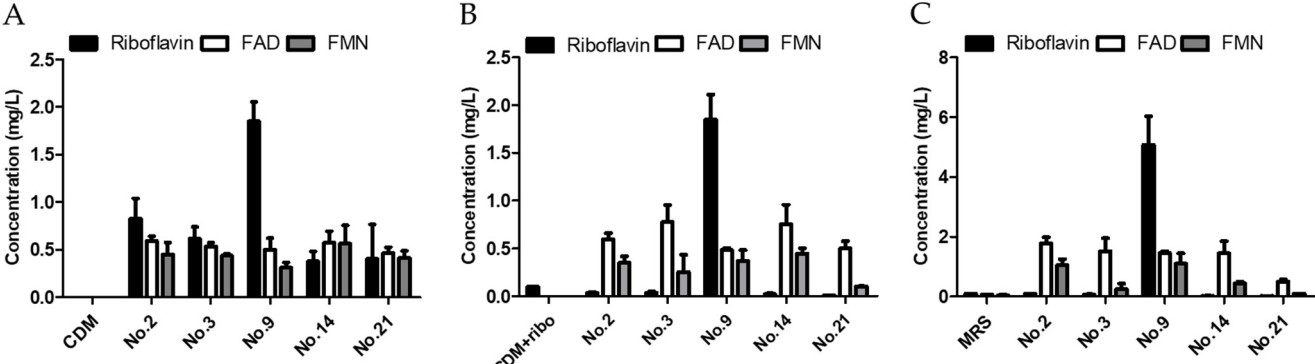

**Figure 2.** Concentration of riboflavin produced by the roseoflavin-resistant LAB strain isolated from Kimchi in (**A**) riboflavin-free chemically defined medium (CDM), (**B**) 0.1 mg/L riboflavin contained CDM, (**C**) MRS, a commercial media for lactic acid bacteria (LAB).

### 3.2. Analysis of Genetic Factors Leading to Riboflavin Overproduction

Microorganisms, including some LAB, produce riboflavin necessary for growth through the rib operon. However, the rib operon does not produce riboflavin more than that necessary for the growth of the microorganism due to the suppression of its activity in the environment in which riboflavin is present [49]. In addition, it is known that where rib operon genes, including ribA, ribB, ribG, and ribH, are absent or incomplete, LAB cannot produce riboflavin and thus cannot grow in the absence of riboflavin [21,50]. One of the *L. plantarum* type strain, NCDO1752, is known to have completely regulated rib operon and riboflavin biosynthesis genes [21,51]. Therefore, mechanisms for excessive riboflavin production by HY7715 were investigated by comparing the riboflavin produced by NCDO1752 and HY7715 and the expression of the riboflavin biosynthesis gene. In addition, the nucleotide sequence of the RFN element at the upstream of rib operon of the two strains was analyzed and compared. HY7715 produced higher riboflavin than NCDO1752 in riboflavin-free CDM and MRS. Moreover, HY7715 produced higher riboflavin when cultured in MRS than in CDM. In contrast, NCDO1752 produced some riboflavin in CDM, with little production in MRS (Figure 3A). The expression of riboflavin biosynthesis genes of HY7715 and NCDO1752 in MRS showed that the mRNA levels of ribA, ribB, ribG, and ribH were significantly higher in HY7715 than in NCDO1752 with ribG and ribH showing remarkably high mRNA levels (Figure 3B). Compared to NCDO1752, HY7715 showed only one nucleotide difference in the RFN region upstream of the rib operon as indicated by the arrow in Figure 3C.

### 3.3. Stability Study of Riboflavin Overproduction Phenotype in HY7715

For strains isolated in this study to have potential industrial utility, the riboflavin overproduction phenotype must remain stable. To confirm this, HY7715 was cultured for 30 generations in riboflavin-free CDM and MRS, and extracellular riboflavin was measured. The riboflavin overproduction phenotype remained stable for at least 30 generations with or without riboflavin (Figure 4).

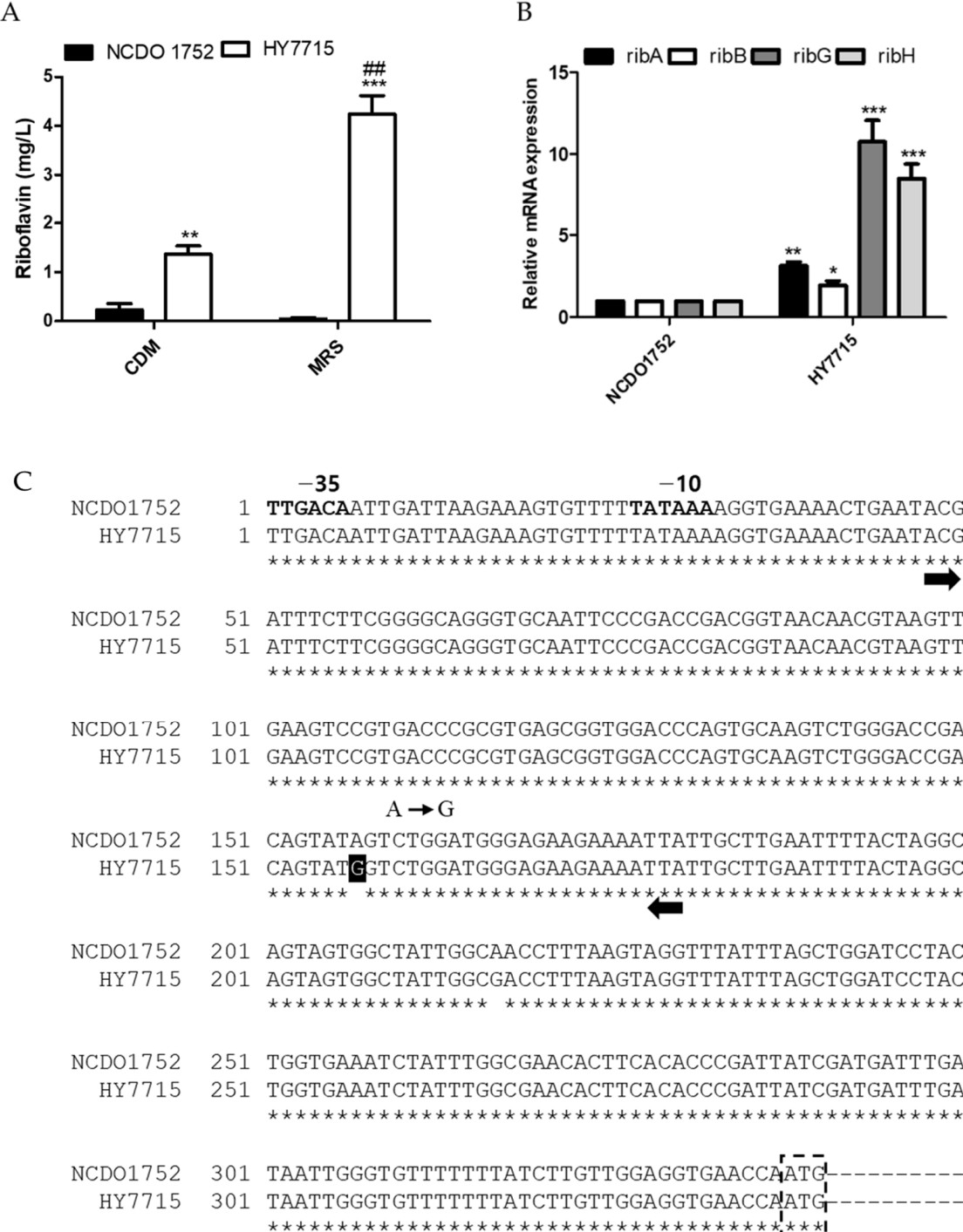

**Figure 3.** Differences between HY7715 and NCDO1752 in (**A**) riboflavin production in CDM and MRS (de Man, Rogosa and Sharpe), (**B**) riboflavin biosynthesis gene (ribA, ribB, ribG, and ribH) expression, and (**C**) rib operon regulatory region (RFN element) sequences. Data are represented as mean ± standard error of means (SEM). * $p < 0.05$, ** $p < 0.01$ and *** $p < 0.001$ compared with NCDO1752. ## $p < 0.001$ compared with CDM-cultured HY7715. The predicted −10 and −35 recognition sequences and ribosome binding sites are shown in bold. RFN elements are indicated by arrows below the sequence. The ribG start codon is indicated by a dashed box.

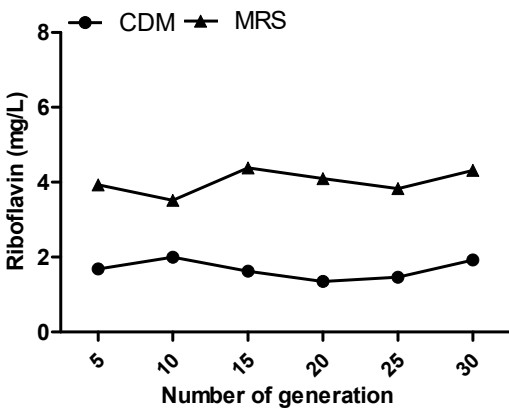

**Figure 4.** Riboflavin produced by HY7715 after successive generations in roseoflavin free CDM and MRS. Riboflavin content in the media was measured by high-performance liquid chromatography (HPLC) every five generations.

This suggested that the riboflavin overproduction phenotype of HY7715 was maintained without any selection pressure. NCDO1752 did not overproduce riboflavin for 30 generations in riboflavin-free CDM and MRS (data not shown).

### 3.4. Optimization of Media Composition and Growth Conditions for Riboflavin Production

Industrial use of probiotic strains requires mass cultivation, for which it is essential to determine the strain-specific low-cost medium and optimal growth conditions [34,35]. Previous studies of riboflavin overproducing strains focused on manufacturing processed foods such as bread, pasta, and soy milk using these strains, and the industrial use of the strain itself was not considered much [18,19,28]. Therefore, determination of optimal media and growth conditions for riboflavin production was performed for the pre-industrial stage of the HY7715. Several carbon and nitrogen sources, and minerals were reviewed with reference to a previous study [34], and the amount of carbon sources, minerals, and trace elements suitable for the growth of the HY7715 were determined (Table 3). For the nitrogen source, skim milk and soluble whey protein, commonly used in probiotic cultivation [35], were used, which did not affect HY7715 growth and riboflavin production (data not shown). HY7715 showed the highest riboflavin production when 3% and 1% of yeast extract and soy peptone (*v/w*), respectively, were used as nitrogen sources (Figure 5A). As growth, temperature, and pH are the main variables in the production of metabolites in bacteria [52,53], riboflavin production of HY7715 in the previously determined optimal medium (Media 2) was measured at various temperatures and pH. The optimum riboflavin production pH of the HY7715 was pH 5.5 and the optimum temperature was 35 °C (Figure 5B,C).

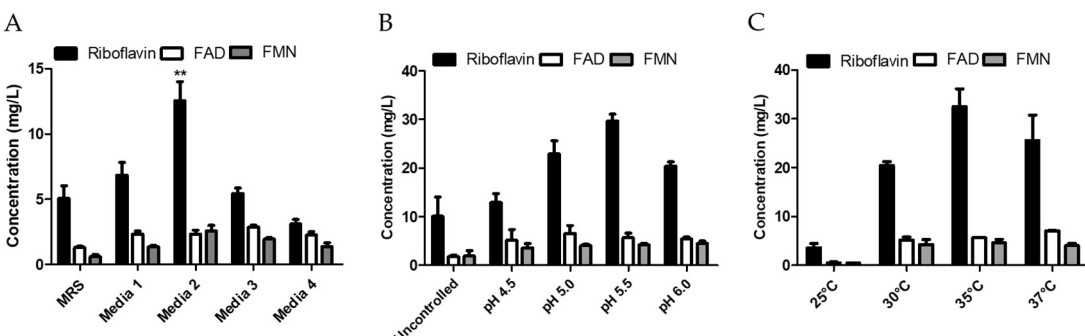

**Figure 5.** Riboflavin production of HY7715 under (**A**) various media composition, (**B**) pH and (**C**) growth temperature conditions. Data are represented as mean ± standard error of means (SEM) of three independent experiments. ** $p < 0.01$ compared with MRS.

### 3.5. Mass Cultivation of HY7715

Stability in large-scale culture is a very important part in the industrial application of useful bacterial strains, such as in probiotics. Accordingly, the stability of HY7715 riboflavin production was verified through 500 L mass fermentation. The composition of the culture medium was the same as that of Media 2, the culture temperature and pH were set to 35 °C and 5.5, respectively. All processes were performed aseptically in hy Probioitics Plant (hy. Co. Ltd.). From measurements of the amount of riboflavin produced by HY7715 for 24 h, results showed that riboflavin increased rapidly up to 21 h, stabilizing thereafter. At 21 and 24 h, riboflavin production was $33.35 \pm 1.34$ mg/L and $34.5 \pm 2.41$ mg/L, respectively. The amount of FAD and FMN did not change significantly over the 24 h (Figure 6A). The number of viable cells was highest at $1.55 \times 10^{10}$ CFU/mL $\pm 1.91 \times 10^{9}$ CFU/mL at 18 h, followed by a slight decrease thereafter (Figure 6B). Interestingly, riboflavin increased even after the viable cells decreased, indicating that HY7715 overproduced riboflavin not only when in the log phase but also in the stationary phase. Therefore, it was confirmed that HY7715 can produce riboflavin without consuming riboflavin in the medium and can stably grow under mass culture conditions.

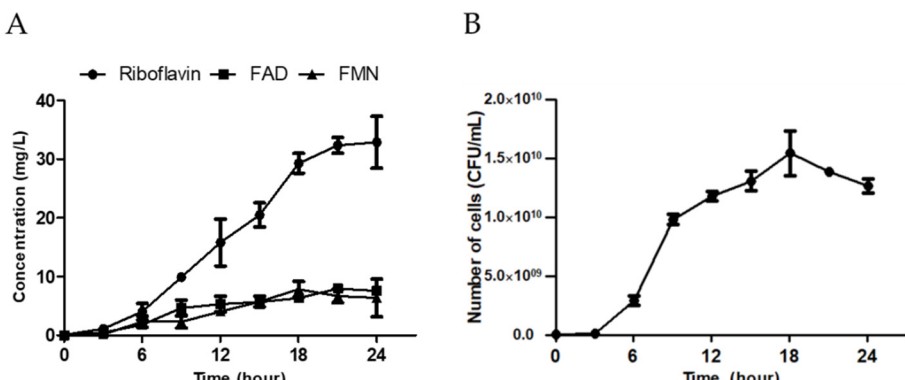

**Figure 6.** Measurement of (**A**) riboflavin production and (**B**) viable cells during mass cultivation of HY7715. Data are represented as mean $\pm$ standard error of means (SEM).

### 3.6. Riboflavin Production of HY7715 in Co-Culture with Caco-2 Cells

To predict whether HY7715 produced riboflavin in the intestinal environment, co-culturing with human intestinal cells, Caco-2, was performed. The riboflavin content of DMEM medium cultured with Caco-2 cells alone was $0.273 \pm 0.015$ mg/L, and when HY7715 was co-cultured, the riboflavin content increased significantly to $0.817 \pm 0.144$ mg/L. However, when NCDO1752 was co-cultured, the riboflavin content in the medium slightly decreased to $0.214 \pm 0.07$ mg/L (Figure 7). During co-culturing, damage to Caco-2 cells, including cell death by LAB, was not observed (data not shown). This observation suggested that HY7715 has the potential to produce riboflavin even under an intestinal environment different than its own optimal growth conditions.

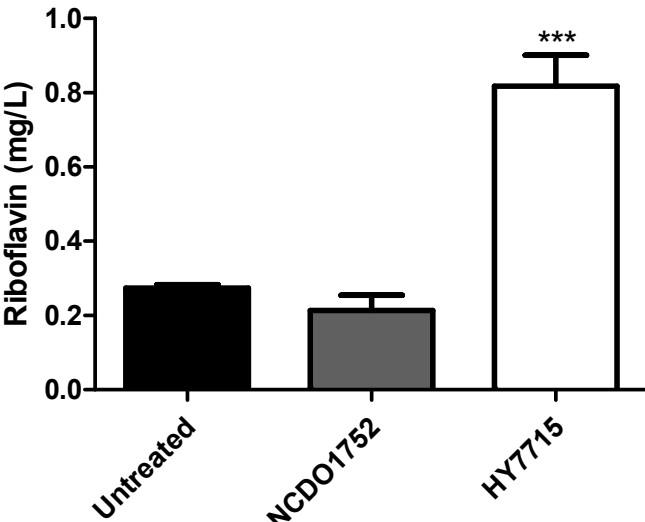

**Figure 7.** Riboflavin production in a co-culture of lactic acid bacteria and Caco-2 cells. HY7715 produced higher riboflavin than NCDO1752, which was used as a control in a co-culture environment. Data are represented as mean ± standard error of means (SEM) of three independent experiments. *** $p < 0.001$ compared with LAB untreated control.

### 3.7. Confirmation of Riboflavin Production of HY7715 In Vivo Using an Animal Model

After inducing riboflavin deficiency by administering HY7715 to mice, riboflavin was measured in plasma and urine to confirm whether HY7715 can continuously produce and supply riboflavin in vivo. The plasma riboflavin concentration of mice not administered with LAB (D) was 0.21 ± 0.64 mg/dL, which was significantly lower than that of the control group (C) 0.65 ± 0.98 mg/dL; however, the plasma riboflavin concentration of mice fed with HY7715 was 0.61 ± 0.04 mg/dL. The plasma riboflavin concentration of mice that received NCDO1752 was 2.9 ± 0.03 mg/dL, which was slightly higher than that of the D group but was the difference was not significant (Figure 8A). The concentration of riboflavin excreted in the urine was 53.49 ± 8.11 mg/dL in the C group, 3.82 ± 1.53 mg/dL in the NCDO1752 group, and 45 ± 5.16 mg/dL in the HY7715 group. The riboflavin concentration in group D urine was 0.75 ± 0.09 mg/dL, which was relatively low compared to other groups (Figure 8B). Therefore, RNA was isolated from cecum contents of mice and the ratio of LAB and *ribG* gene levels were analyzed. From the analysis of the LAB ratio in the cecum, using LAB-specific 16S rRNA primers, it was confirmed that the LAB increased by about four times, as compared to the C group, in both NCDO1752 and HY7715 groups. Group D showed similar LAB ratio to group C (Figure 8C). The mRNA levels of the ribG were about 150 times higher in the HY7715 group than in group C or D and 14 times higher than in the NDCO1752 group (Figure 8D). There were no significant differences in body weight and food intake between groups during the experiment (Figure S2). This indicated the presence of bacterial species that could produce riboflavin in the intestine of HY7715 group, which indirectly showed that HY7715 is settling in the intestine.

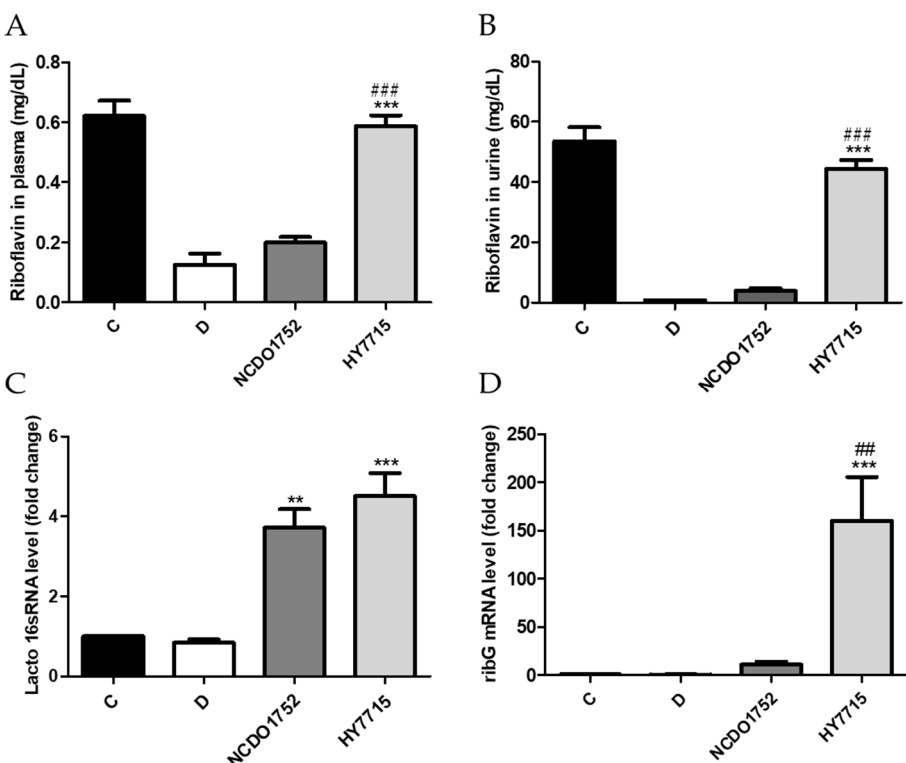

**Figure 8.** Riboflavin concentration in (**A**) plasma and (**B**) urine. (**C**) Relative LAB ratio of cecum contents (Relative LAB-specific 16S rRNA levels). (**D**) Relative ribG mRNA levels in cecum contents. Data are represented as mean ± standard error of means (SEM). ** $p < 0.01$ and *** $p < 0.001$ compared with D group. ## $p < 0.01$ and ### $p < 0.001$ compared with NCDO1752 group.

### 3.8. Probiotic Properties of HY7715

After measuring digestive tract viability of HY7715 through a simulated GIT model, HY7715 showed significantly higher intestinal viability than NCDO1752 (Figure 9A). In addition, in an in vitro test measuring the intestinal cell adhesion ability of LAB, about 89 HY7715 adhered to Caco-2 cell for 2 h, whereas only about 22 NCDO1752 adhered to Caco-2 cell (Figure 9B). About 68 *L. plantarum 299V*, well known as a probiotic strain [41] and the same species as HY7715, attached to Caco-2 cell under the same conditions (data not shown). Therefore, HY7715 had high viability in the digestive tract and an excellent intestinal adhesion, suggesting that it has the potential of probiotics.

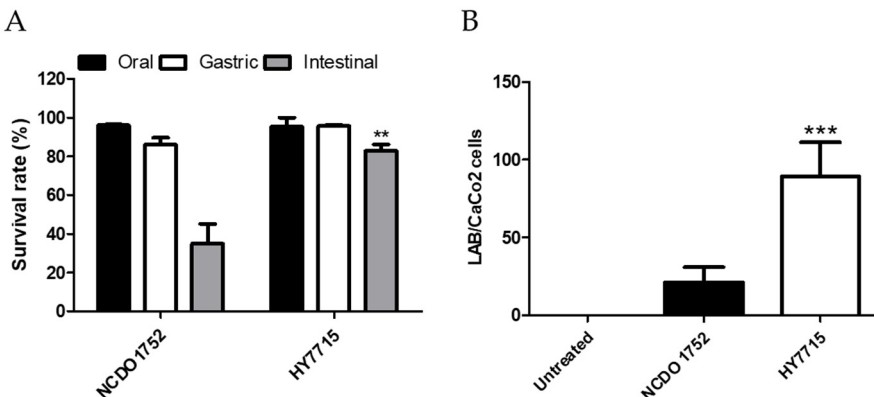

**Figure 9.** (**A**) Survival rate during simulated gastrointestinal digestion of HY7715 and NCDO1752. (**B**) The number of LAB attached to one Caco-2 cell under co-culture conditions. "LAB/Caco-2 cells" indicate colony forming units (CFU) per cell. Data are represented as mean ± standard error of means (SEM) of three independent experiments. ** $p < 0.01$ and *** $p < 0.001$ compared with NCDO1752.

From conducting an antibiotic resistance test, according to the standard test method [35], HY7715 was found to be lower than the antibiotic resistance suggested by EFSA (Table S3). The strain did not show β-hemolytic reaction in the test using blood agar plate (Figure S3). In addition, after confirming the existence of potential virulence genes through whole-genome analysis, it was verified that HY7715 did not have any commonly known virulence genes (data not shown).

## 4. Discussion

Lactic acid bacteria (LAB) are a group of microorganisms widely used in the industry for the production of fermented foods. In addition, LAB are used as probiotics in various functional products due to their various properties with health benefits [24]. One of the health benefits of LAB includes supplying vitamins to the host through *de novo* synthesis [50,54]. Vitamin B2, also known as riboflavin, is not synthesized in the human body and must be supplied through ingestion [1]. Accordingly, many researchers have discovered LAB strains that overproduce riboflavin and conducted research to manufacture riboflavin-enriched foods using these strains [18–21]. Roseoflavin is an analog of riboflavin, which is toxic to microorganism as it cannot be used for the synthesis of FAN and FMN, the co-enzyme forms of riboflavin. Therefore, it has been reported that roseoflavin can be used to increase riboflavin production in microorganisms by inducing mutations in RFN element upstream of the rip operon [20,21]. Thus, LAB that are inherently resistant to roseoflavin have the potential to overproduce riboflavin. In this study, LAB that overproduces riboflavin was isolated from Kimchi, a known fermented vegetable with high vitamin B2 content. LAB strains overproducing riboflavin were selected through riboflavin-free CDM plates containing roseoflavin. Of the LAB isolated from 26 Kimchi samples, 46 strains were able to grow without riboflavin, with 18 strains producing riboflavin at a detectable level in CDM (Table S1). There were five strains capable of forming colonies even in an environment in which roseoflavin was present at a high concentration (150 μg/mL), and only one strain produced very high riboflavin regardless of the presence or absence of riboflavin (Figure 2). The species of the finally selected strain was *L. plantarum*, which was named HY7715. HY7715 produced significantly higher riboflavin than the *L. plantarum* type strain NCDO1752, which is known to grow without riboflavin (Figure 3A). This riboflavin-overproducing ability of HY7715 was predicted to be due to high *ribA*, *ribB*, *ribC*, *ribH*, and *ribG* gene expression (Figure 3B). Interestingly, HY7715 and NCDO1752 showed only one nucleotide difference in the rib operon regulatory region, which was consistent with previous studies [19,21] that a single nucleotide mutation alters the operon regulation ability, resulting in increased riboflavin production (Figure 3C). The riboflavin overproduction ability of HY7715 was maintained for at least 30 days even in the absence of selective pressure (Figure 4). Moreover, the riboflavin-overproducing ability of HY7715 was maintained even in mass culture, a prerequisite for industrialization (Figure 6A). This suggested that HY7715 could become an industrially valuable LAB in the area of riboflavin biosynthesis.

Riboflavin deficiency often occurs when nutritional intake is unbalanced because riboflavin is not synthesized in the body and the remaining riboflavin that has not been converted to a coenzyme is excreted in the urine [2,5,8]. There have been previous studies to improve riboflavin deficiency with fermented soymilk using riboflavin overproducing LAB strains [16]. EGRAC, one of the indicators of riboflavin deficiency, was found to be improved, but riboflavin uptake and excretion in vivo were not confirmed in that study. Therefore, in this study, it was confirmed that the intake of HY7715, a new riboflavin-overproducing strain, can increase riboflavin levels in-vivo. Mice that were pre-supplied with HY7715 showed high levels of riboflavin in plasma and urine even when riboflavin deficiency was induced through nutrient restriction (Figure 8A,B). The fact that riboflavin was excreted in the urine even when riboflavin was not supplied suggested that HY7715 produces and supplies enough riboflavin in the mouse body. In addition, in mice fed with HY7715, even after 4 days of cessation of HY7715 feeding, the LAB ratio in the cecum was four times higher than in the control, and the mRNA level of *ribG*, a riboflavin biosynthesis

gene, was approximately 150 times higher (Figure 8C,D). In previous experiments, it was confirmed that HY7715 expresses a relatively high *ribG* gene. In addition, Mohedano et al. suggested that the level of *ribG* gene in intestinal microorganisms may be an indicator of riboflavin production in intestinal microorganisms [55]. These results suggested that HY7715 can colonized in the mouse intestine for at least 3 days or more to continuously produce riboflavin.

Although many LAB strains are used as probiotics, not all LAB are probiotics, and only strains that have secured gastrointestinal stability and biological safety can be proposed as probiotics [35,56]. The ability to attach to intestinal epithelium and survive in the digestive tract is one of the main criteria for the selection of probiotic strains [39]. Probiotics taken by mouth reach the intestine through a stressful environment such as saliva, gastric juice, and bile salt. Thus, the high digestive viability and ability to attach to the intestine are consistent with the effectiveness of the probiotic strain [39,41]. The simulated GIT showed that HY7715 had a very high viability, especially in the intestinal environment, resulting in better digestive stability than the type strain (Figure 9A). HY7715 adhered well to human intestinal epithelial cells, Caco-2, and did not lose the ability to produce riboflavin even during co-culture with Caco-2 (Figures 7 and 9B).

Antibiotic resistance, hemolytic activity and virulence factors are very important indicators to confirm the safety of probiotics [24,35,41]. No matter how useful the LAB strain is, it cannot be used as a probiotic if its antibiotic resistance is higher than standard or it produces toxic substances. In this study, it was confirmed that HY7715 had antibiotic resistance below the standard suggested by EFSA and did not cause hemolysis in blood agar plate (Table S3 and Figure S3). Moreover, it was confirmed that there were no virulence genes in HY7715, through whole genome sequence analysis (data not shown). Therefore, HY7715 could be considered a valuable LAB strain with high probiotic potential.

## 5. Conclusions

This study isolated a novel LAB that overproduces vitamin B2 from fermented Kimchi and validated its potential as a probiotic. HY7715 produced the highest amounts of riboflavin among isolated LAB strains, and these properties were maintained throughout the generations. HY7715 produced higher riboflavin than any previously reported LAB strains [18–20,24,28] under optimal growth conditions and did not consume riboflavin in the medium; the strain was also able to produce riboflavin in Caco-2 co-culture, an in vitro model of the intestinal environment. Furthermore, high levels of riboflavin were detected in plasma and urine in animals fed HY7715 even when riboflavin deficiency was induced through nutritional restriction. This suggested that HY7715 may settle in the intestine and continue to produce riboflavin. HY7715 had a high digestive survival rate, showed antibiotic resistance below the cutoff MIC value suggested by EFSA, and did not cause hemolysis. Therefore, HY7715 could be used as a beneficial probiotic strain in human and animal applications, suggesting that HY7715 is a new alternative to riboflavin supplementation that could help address riboflavin deficiency.

**Supplementary Materials:** The following are available online at https://www.mdpi.com/article/10.3390/app11135765/s1, Table S1: Confirmation of riboflavin production in CMD of LAB isolated from Kimchi, Table S2: Composition of AIN-93G and AIN-93G_RD diet, Table S3: Antibiotic resistance and minimum inhibitory concentrations (MIC) Figure S1: Example of qualita-tive detection of riboflavin produced by change of color from white to yellow in riboflavin-free chemically defended media (CDM), Figure S2: Changes in (A) body weight and (B) food intake in the experimental groups, Figure S3: Confirmation of β-hemolytic reaction of HY7715 in blood agar plate.

**Author Contributions:** Conceptualization, J.-Y.K., J.-J.S. and J.-L.L.; methodology, J.-Y.K.; software, J.-Y.K., M.-S.Y.; validation, J.-Y.K. and J.-H.L.; formal analysis, J.-Y.K.; investigation, J.-Y.K., M.-S.Y., E.-J.C. and J.-H.L.; writing—original draft preparation, J.-Y.K.; writing—review and editing, K.H.; visualization, J.-Y.K. All authors have read and agreed to the published version of the manuscript.

**Funding:** This research received no external funding.

**Institutional Review Board Statement:** Animal experiments were reviewed and approved by the Institutional Animal Care and Use Committee of hy Co., Ltd.

**Informed Consent Statement:** Not applicable.

**Data Availability Statement:** The data presented in this study are available in the article.

**Conflicts of Interest:** The authors declare no conflict of interest.

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
