# Peer review of "Probiotic Potential of a Novel Vitamin B2-Overproducing Lactobacillus plantarum Strain, HY7715, Isolated from Kimchi"

_applsci, doi:10.3390/app11135765_

Round 1

Reviewer 1 Report

The manuscript entitled “Probiotic Potential of a Novel Vitamin B2-overproducing Lactobacillus plantarum strain, HY7715, Isolated from Kimchi” submitted for revision in the Applied Sciences had been positively reviewed with some minor modifications.

 This is an interesting manuscript concerning to isolated a  new  LAB that overproduces vitamin  B2  from the fermented kimchi, to optimize vitamin  B2  production,  and to verify its potential as a probiotic. The experiment is extensive and has been conducted in various in vitro and in vivo models. Interesting results were obtained, which were properly discussed by the authors. The research methods were properly selected and correct conclusions were drawn.

The Authors suggest there, HY7715 could be used as a beneficial probiotic  strain  in  human  and  animal  applications,  and could  be  a  new alternative  to  riboflavin  supplementation  that  could  help  address  riboflavin deficiency.

The review is positive. Therefore, I propose to accept this paper for publication after minor amendments.

  1. Title: there is a mistake in the title, it should be: Lactobacillus not Lac-tobacillus.
  2. Abstract: There are no precisely defined research objectives.
  3. The keywords must be changed. Keywords should not be the same as in the title of the manuscript. All keywords are included in the title: “Vitamin B2; Probiotics; Lactobacillus plantarum; Kimchi”.
  4. In the in vivo experiment, the composition of the diets should be given (in table).
  5. Also in the in vivo experiment nutritional parameters and nutritional status (amount of feed consumed, weight gain) should also be given.
  6. The method of determination riboflavin levels in plasma should be described, and also: cecum contents. Not only: “… were analyzed by HPLC...” “…performing real-time PCR after RNA isolation...”
  7. In the text (lines 284-286), described as: “In addition, strain 9  produced  06±2.05  mg/L  of  riboflavinin  MRS  broth,  which  was  much  higher than that of the other…

Were there significant differences between the results (Figure 2)? If so, please mark it on the chart. If there were no significant differences, it cannot be written that it was higher ...

  1. Figure 3 is too small and illegible. You have to correct it.
  2. Figure 7 - I believe that it is not necessary to mark *** and ###. It is enough if it is only marked *** because this already suggests that there is a significant difference between HY7715  and LAB untreated control and also NCDO1752.
  3. Line 459 - which means (Table S1)?
  4. References - newer literature should be added.

Author Response

Dear reviewer

Thanks for your great advice.

Response to Reviewer 1 Comments

Point 1: Title: there is a mistake in the title, it should be: Lactobacillus not Lac-tobacillus.

Response 1: Thanks for the advice. It was a typo and has been corrected in the text.

Point 2: Abstract: There are no precisely defined research objectives.

Response 2: “The purpose of this study was to isolate novel lactic acid bacteria that overproduce vitamin B2 and to validate their potential as probiotics.” We included these in the abstract draft, but ultimately deleted it because the abstract was limited to 200 characters. If there are no editorial issues, I will include the content.

Point 3: The keywords must be changed. Keywords should not be the same as in the title of the manuscript. All keywords are included in the title: “Vitamin B2; Probiotics; Lactobacillus plantarum; Kimchi”.

Response 3: Thank you very much for your advice. I've edited the keywords based on your advice.

Point 4: In the in vivo experiment, the composition of the diets should be given (in table).

Response 4: The composition of dietary ingredients has been added to the Supplementary Material section.

Point 5: Also in the in vivo experiment nutritional parameters and nutritional status (amount of feed consumed, weight gain) should also be given.

Response 5: Since there was no significant difference in nutritional parameters and nutritional status in this experiment. But, as you mentioned, the sentence added as “There were no significant differences in body weight and food intake between groups during the experiment”. In addition, the dietary intake and weight measurement results were added to the Supplementary Material section.

Point 6: The method of determination riboflavin levels in plasma should be described, and also: cecum contents. Not only: “… were analyzed by HPLC...” “…performing real-time PCR after RNA isolation...”

Response 6: Methods for determining plasma riboflavin level was described in Method section 2.3 ‘Quantitative analysis of vitamin B2’. cDNA synthesis and real-time PCR methods were described in section 2.4 'Expression analysis of riboflavin biosynthesis genes' and the following sentence was added to section 2.10 'Total RNA was isolated using the AllPrep PowerFecal RNA kit (QIAGEN, German -town, MD, United States), and cDNA synthesis and real-time PCR were performed as described in section 2.4'.

Point 7: In the text (lines 284-286), described as: “In addition, strain 9  produced  5.06±2.05  mg/L  of  riboflavinin  MRS  broth,  which  was  much  higher than that of the other…”
Were there significant differences between the results (Figure 2)? If so, please mark it on the chart. If there were no significant differences, it cannot be written that it was higher …

Response 7: Thank you for your advice. The riboflavin production levels of strain 9 showed a significant difference between the results (Figure 2). However, it is not necessary to show the difference between the results in Figure 2, so the above sentence has been modified to read: ' In addition, strain 9 produced 5.06±2.05 mg/L of riboflavin in MRS broth, while the other strains produced little riboflavin in MRS broth (Figure 2C)'.

Point 8: Figure 3 is too small and illegible. You have to correct it.

Response 8: Figure 3 has been resized according to your advice.

Point 9: Figure 7 - I believe that it is not necessary to mark *** and ###. It is enough if it is only marked *** because this already suggests that there is a significant difference between HY7715  and LAB untreated control and also NCDO1752.

Response 9: Thank you for your comments. The ### mark has been removed from the Figure 7.

Point 10: Line 459 - which means (Table S1)?

Response 10: Table S1 refers to Table 1 of Supplementary Materials. Table S1 contains a list of 46 LAB strains which can grow in riboflavin-free CDM. Table S1 also contains information on the riboflavin production level and roseoflavin resistance of each strain.

Point 11: References - newer literature should be added.

Response 11: Some of recent publications were included in the References section. Please check the list below. 

10. Mosegaard, S.; Dipace, G.; Bross, P.; Carlsen, J.; Gregersen, N.; Olsen, R.K.J. Riboflavin Deficiency-Implications for General Human Health and Inborn Errors of Metabolism. Int. J. Mol. Sci. 2020, 21 (11), doi:10.3390/ijms21113847.
20. Ge, Y.Y.; Zhang, J.R.; Corke, H.; Gan, R.Y. Screening and Spontaneous Mutation of Pickle-Derived Lactobacillus plantarum with Overproduction of Riboflavin, Related Mechanism, and Food Application. Foods 2020, 9 (1), doi:10.3390/foods9010088.
25. Daliri, E.B.; Ofosu, F.K.; Chelliah, R.; Lee, B.H.; Oh, D.H. Challenges and Perspective in Integrated Multi-Omics in Gut Microbiota Studies. Biomolecules 2021, 11 (2), doi:10.3390/biom11020300.
28. Zhu, Y.Y.; Thakur, K.; Feng, J.Y.; Cai, J.S.; Zhang, J.G.; Hu, F.; Russo, P.; Spano, G.; Wei, Z.J. Riboflavin-Overproducing Lactobacilli for the Enrichment of Fermented Soymilk: Insights into Improved Nutritional and Functional Attributes. Appl. Microbiol. Biotechnol. 2020, 104 (13), 5759–5772, doi:10.1007/s00253-020-10649-1.
33. Oleksy-Sobczak, M.; Klewicka, E. Optimization of Media Composition to Maximize the Yield of Exopolysaccharides Production by Lactobacillus rhamnosus Strains. Probiotics Antimicrob. Proteins 2020, 12 (2), 774–783, doi:10.1007/s12602-019-09581-2.
39. Jung, S.H.; Hong, D.K.; Bang, S.-J.; Heo, K.; Sim, J.-J.; Lee, J.-L. The Functional Properties of Lactobacillus casei HY2782 Are Affected by the Fermentation Time. Appl. Sci. 2021, 11 (6), doi:10.3390/app11062481.
41. Arellano, K.; Vazquez, J.; Park, H.; Lim, J.; Ji, Y.; Kang, H.J.; Cho, D.; Jeong, H.W.; Holzapfel, W.H. Safety Evaluation and Whole-Genome Annotation of Lactobacillus plantarum Strains from Different Sources with Special Focus on Isolates from Green Tea. Probiotics Antimicrob. Proteins 2020, 12 (3), 1057–1070, doi:10.1007/s12602-019-09620-y.
51. Noman, A.E.; Al-Barha, N.S.; Sharaf, A.M.; Al-Maqtari, Q.A.; Mohedein, A.; Mohammed, H.H.H.; Chen, F. A Novel Strain of Acetic Acid Bacteria Gluconobacter oxydans FBFS97 Involved in Riboflavin Production. Sci. Rep. 2020, 10 (1), 13527, doi:10.1038/s41598-020-70404-4.

Reviewer 2 Report

This is interesting paper and would like to support the publication of these results. Authors have well planned and executed the research plan. However, some adjustments need to be done:

Ln3: Please, correct name of Lactobacillus. Maybe authors will consider applying new taxonomy of the lactobacilli?

Ln60-64: Please, try to present this section better.

Ln69: To keep samples at -80C is not very good idea. Since the system do not contain cryo protector/s, most probably at that temperature most of the bacterial representatives will be dead.

Ln71: M17 lactose or M17 glucose?

Ln72: ..Sparks, MD, USA).

Isolation process was at what temperature, anaerobic, or aerobic? Please, be more specific.

Ln112-123: This section need a bit more details.

Ln126: L.  plantarum

Ln224, 227, etc: CaCl2 - 2 needs to be in index.

Please, only first time you need to provide full address of provider, in second case, only name of the company is sufficient.

Ln260: Lactobacillus (L.) plantarum this needs to be introduced much more earlier in the manuscript.

Ln266: 16S rRNA

Authors have any evidences that different isolated L. plantarum are in fact different strains? Any repPCR, RAPD-PCR? Any evidences that they are not copy of same strain, isolated in multiple occasions?

Ln466: Gene names needs to be in italics.

Author Response

Dear reviewer

Thank you for your excellent review.

Our manuscript has been modified to reflect your advice.

Please check the response below.

Thanks again for your consideration.

Response to Reviewer 2 Comments

Point 1: Ln3: Please, correct name of Lactobacillus. Maybe authors will consider applying new taxonomy of the lactobacilli?

Response 1: Thanks for the advice. It was a typo and has been corrected in the text.

Point 2: Ln60-64: Please, try to present this section better.

Response 2: We have edited the content according to your advice.

Point 3: Ln69: To keep samples at -80C is not very good idea. Since the system do not contain cryo protector/s, most probably at that temperature most of the bacterial representatives will be dead.

Response 3: Thanks for your comment. We used glycerin as a cryoprotectant. These contents were added to the Method section.

Point 4: Ln71: M17 lactose or M17 glucose?

Response 4: It was M17 glucose. The text has been edited.

Point 5: Ln72: ..Sparks, MD, USA).

Response 5: Do you mean to change 'United States' to 'USA'? With that in mind, I replaced all 'United States' to 'USA'.

Point 6: Isolation process was at what temperature, anaerobic, or aerobic? Please, be more specific.

Response 6: Added the following sentence to the method section: 'The plates were incubated for 48 h in an anaerobic chamber (Coy Laboratory Products, Grass Lake, MI, USA) at 37 °C.'.

Point 7: Ln112-123: This section need a bit more details.

Response 7: We have edited the content according to your comments.

Point 8: Ln126: L.  plantarum

Response 8: The typo has been corrected in the text.

Point 9: Ln224, 227, etc: CaCl2 - 2 needs to be in index.

Response 9: I corrected the typo according to your advice.

Point 10: Please, only first time you need to provide full address of provider, in second case, only name of the company is sufficient.

Response 10: All corrections have been made to reflect your comments.

Point 11: Ln260: Lactobacillus (L.) plantarum this needs to be introduced much more earlier in the manuscript.

Response 11: It has been modified to reflect your advice.

Point 12: Ln266: 16S rRNA

Response 12: 16S RNA was edited to 16S rRNA 

Point 13: Authors have any evidences that different isolated L. plantarum are in fact different strains? Any repPCR, RAPD-PCR? Any evidences that they are not copy of same strain, isolated in multiple occasions?

Response 13: We isolated LAB from 50 different kimchi samples collected from 26 regions across the country. The kimchi samples we collected were made by traditional methods in different regions, so it is unlikely that the same LAB strain was detected. In addition, full genome sequence analysis was performed in 4 Lactobacillus plantarum (strain No. 2, No. 3, No. 9, No. 14) out of the final selected strains, and the result showed that those were not identical.

Point 14: Ln466: Gene names needs to be in italics.

Response 14: Thanks for the advice. It has been corrected.

This manuscript is a resubmission of an earlier submission. The following is a list of the peer review reports and author responses from that submission.